# Evidence Accumulates: Patients with Ascending Aneurysms Are Strongly Protected from Atherosclerotic Disease

**DOI:** 10.3390/ijms242115640

**Published:** 2023-10-27

**Authors:** Christina Waldron, Mohammad A. Zafar, Bulat A. Ziganshin, Gabe Weininger, Nimrat Grewal, John A. Elefteriades

**Affiliations:** 1Aortic Institute at Yale-New Haven Hospital, Yale University School of Medicine, New Haven, CT 06519, USA; christina.waldron@yale.edu (C.W.); mohammad.zafar@yale.edu (M.A.Z.); bulat.ziganshin@yale.edu (B.A.Z.);; 2Department of Cardiovascular and Endovascular Surgery, Kazan State Medical University, 420012 Kazan, Russia; 3Department of Cardiothoracic Surgery, Amsterdam University Medical Center, 1105 AZ Amsterdam, The Netherlands; n.grewal@amsterdamumc.nl

**Keywords:** atherosclerosis, thoracic aortic aneurysm, abdominal aneurysm, intimal medial thickness, vascular calcification, myocardial infarction

## Abstract

Ascending thoracic aortic aneurysms may be fatal upon rupture or dissection and remain a leading cause of death in the developed world. Understanding the pathophysiology of the development of ascending thoracic aortic aneurysms may help reduce the morbidity and mortality of this disease. In this review, we will discuss our current understanding of the protective relationship between ascending thoracic aortic aneurysms and the development of atherosclerosis, including decreased carotid intima–media thickness, low-density lipoprotein levels, coronary and aortic calcification, and incidence of myocardial infarction. We also propose several possible mechanisms driving this relationship, including matrix metalloproteinase proteins and transforming growth factor-β.

## 1. Introduction

It is well understood that the pathogenesis of abdominal aortic aneurysms (AAAs) derives from atherosclerotic risk factors and plaque development [1,2,3,4]. However, as we shall see, there is a growing body of evidence suggesting that ascending thoracic aortic aneurysms (ATAAs) and descending thoracic/aortic aneurysms (DTAAs) are distinct disease entities, with different genetic, embryologic, anatomic, histological, and biochemical characteristics in both health and disease.

Over the course of several decades, operating on thousands of ATAAs, our team noticed that ATAA patients had soft, supple femoral arteries and ascending aortas free from atherosclerosis, calcification, and thrombus. Subsequent clinical studies by our team demonstrated that ATAA patients had lower carotid intima-media thickness and total body calcium scores (vs. controls), inversely correlated low-density lipoprotein (LDL) levels, and near-total protection from myocardial infarction. Recent histologic evidence also suggests a dearth of atherosclerosis in ATAA patients.

Our work and that of many other teams has demonstrated that many ascending aortic aneurysms are often familial in origin and genetically triggered [5,6]. Among the minority of patients without genetic etiology, inflammation and infection are known, non-atherosclerotic triggers [7,8,9,10].

Understanding the differences between this ‘silver lining’ protective mechanism of ATAAs and the pro-atherosclerotic progression of AAAs is critical for the future development of novel therapies to protect against degenerative processes in both segments of the thoracic aorta, ascending and descending. This review summarizes our current understanding of the divergence between the ascending thoracic aorta and abdominal aorta as distinct disease entities—and the consequent influence on the frequency of atherosclerosis in each segment, ascending and descending.

## 2. Embryology of the Aorta

There is a growing body of evidence that the aorta can be viewed as two distinct organs, (1) ascending and (2) thoracoabdominal (descending and abdominal), separated by the ligamentum arteriosum [11,12] (Figure 1). It has been suggested that these changes begin embryologically [13], with ATAAs ultimately demonstrating pro-aneurysmal and anti-atherosclerotic tendencies, whereas the thoracoabdominal aorta develops highly atherosclerotic aneurysms. The vascular smooth muscle cells (VSMCs), which are the predominant cell type in the middle aortic layer, have different embryonic origins, and the migrations of these VSMCs are also site specific [14]. During embryological development, VSMCs in the aortic root are neural crest- and secondary heart field-derived. In the ascending aorta and arch, up to the ligamentum arteriosum, the VSMCs are derived from neural crest cells, and in contrast, the VSMCs of the descending thoracic and abdominal aorta are derived from the paraxial mesoderm [13]. The differences in embryologic cellular origin have been found to contribute to a differentiation defect of VSMCs in aneurysm formation, resulting in a structurally distinct aortic media composition with pathological VSMC responsiveness to growth factors and an excessive extracellular matrix accumulation [15]. This fundamental difference in tissue of origin therefore translates into marked differences in the character of aneurysms in the distinct aortic segments.

## 3. Our Investigations

Having noted that ascending aortic aneurysms are smooth in contour, non-atherosclerotic, and non-thrombus containing, while thoracoabdominal aortic aneurysms are irregular in contour, highly atherosclerotic, and full of thrombus, we embarked on a series of studies to investigate these differences further.

### 3.1. Intimal Medial Thickness

An early marker of atherosclerosis is carotid intima–media thickness (IMT), which can be measured as the distance between the lumen-intimal and the medial-adventitial interface using non-invasive B-mode ultrasound. IMT has been shown to be reliable and reproducible across a wide variety of populations [16,17,18,19,20,21]. Importantly, thicker IMT and/or carotid plaques indicate atherosclerosis and predict cardiovascular mortality [22,23,24]. Specifically, increased IMT is a strong predictor of stroke, coronary artery disease, and myocardial infarction [18,19,20,21,25,26,27,28,29,30]. As arterial wall injury and inflammation progress during the subclinical stages of atherosclerosis, the carotid wall measurably increases [31]. One meta-analysis has shown that for every 0.1 mm increase in IMT, the risk of myocardial necrosis and stroke increase by 10–15% and 13–18%, respectively [32]. Other large studies replicate these trends, and the American College of Cardiology Foundation/American Heart Association include IMT in their guidelines for predicting cardiovascular events [33,34,35,36,37]. Although some recent studies have questioned its utility, carotid IMT remains an established and frequently used imaging study for the current extent and future progression of atherosclerosis [38,39].

In our prior study led by Dr. Hung et al., we investigated this relationship by comparing the carotid IMT between ascending aneurysm patients and age-matched controls [40]. In this study, the carotid IMT of 52 patients with ascending thoracic aortic aneurysms and 29 controls were measured using ultrasound, with 6 IMT measurements per patient. None of these patients had Marfan Syndrome or any other connective tissue disease. To control for age, environment, ethnicity, and lifestyle, the spouses of the patients were used for controls. The experimental group consisted of statistically significantly more male patients than the control group, with 75% vs. 21% male patients. Critically, we found that patients with ATAAs had 0.131 mm lower carotid IMT values than controls after controlling for common risk factors of atherosclerosis, including BMI, age, gender, family history, smoking, dyslipidemia, race, diabetes, and hypertension. Importantly, a 0.1 mm decrease in IMT has a significant impact on the risk of cardiovascular comorbidity, corresponding to a 13–18% decrease in the risk of stroke and a 10–15% decrease in the risk of myocardial infarction [32]. The average IMT in patients with ATAAs had an IMT of 0.5 mm, compared to 0.6 mm in controls (a normal value for this group’s average age of 59) (Figure 2). Not only was IMT not higher in the ascending aneurysm patients, but it was statistically significantly lower. Age was the only statistically significant risk factor that increased IMT, with IMT increasing by 0.005 mm for every 1-year increase in age (Figure 3). Other non-significant factors that increased IMT included male gender, dyslipidemia, family history, diabetes, and hypertension, all of which increased the IMT value by 0.046, 0.012, 0.032, 0.027, and 0.033 mm, respectively. As the aneurysm group was composed almost entirely of males, the lower IMT in the aneurysm group becomes even more impressive. These findings strongly support the protective effect of ATAAs against the development of atherosclerosis.

### 3.2. Lipid Profiles

High plasma levels of low-density lipoprotein (LDL) are strongly associated with atherosclerotic processes, including coronary artery disease, stroke, and peripheral artery disease. LDL has been previously documented in several studies to be elevated in patients with AAAs (Figure 4) [41,42,43,44]. As a meaningful marker and risk factor for vascular disease, LDL may provide further insight into the nature of the relationship between ascending thoracic aortic aneurysms and atherosclerosis [45,46,47,48,49,50].

Our study led by Dr. Weininger et al. investigated this relationship by comparing low-density lipoprotein (LDL) levels between patients with ascending thoracic aortic aneurysms and a control population after propensity score matching [51]. None of these patients had Marfan Syndrome. Using a restricted cubic spline model, a sigmoidal relationship between LDL levels and odds of ascending thoracic aortic aneurysm was discovered. Strikingly, we found that lower LDL levels (75 mg/dL) were associated with increased odds of ascending thoracic aortic aneurysms (OR 1.21) whereas elevated LDL levels (150 and 200 mg/dL) were associated with decreased odds of ascending thoracic aortic aneurysms (OR 0.62 and 0.29, respectively) (Figure 5). This observed inverse correlation between LDL levels and ATAAs is a critical addition to the body of evidence showing disparate disease processes for ATAAs and abdominal aneurysms, as LDL is a known, powerful driver of atherosclerosis [52,53,54].

An earlier study replicates these findings, showing an independent association between LDL levels and abdominal aortic aneurysms and no discernible relationship between LDL levels and both patients with ATAAs and controls [43]. Other papers have looked at high-density lipoprotein (HDL) and found that patients with ATAAs had higher levels of HDL and lower incidence of CAD [55]. Together, these findings provide substantial evidence of thoracic and descending/abdominal aortas as distinct organs with separate pathophysiological processes.

This information raised the possibility that the lower lipid levels in ATAA patients may underlie the protection from atherosclerosis.

### 3.3. Coronary Artery and Aortic Calcification

Calcification of the coronary arteries and the aorta is a late manifestation of atherosclerosis, with higher degrees of calcification corresponding with a greater extent of atherosclerosis [56,57,58,59,60,61,62,63,64,65,66,67,68,69,70,71,72,73]. Computed tomography (CT) is a noninvasive, reproducible method for detecting calcified plaques throughout the vasculature [74,75,76]. Increased calcification of the coronary arteries is associated with an increased risk of cardiovascular events [77,78], with one study showing that calcification of the coronary arteries is associated with an 8.7-fold increase in cardiac events and a 4.2-fold increase in death or myocardial infarction (MI) [79].

Our study led by Dr. Achneck et al. investigated this relationship in aortic aneurysm disease by comparing the degree of coronary and aortic calcification between patients with annuloaortic ectasia and/or type A dissections versus control patients [80]. This study included 31 patients with aortic root dilatation and 33 patients with type A dissection, compared to 128 controls. None of these patients had Marfan Syndrome. The degree of calcification in the coronary arteries and aorta was quantified by a CT scan. The three coronary arteries were analyzed separately, and the aorta was divided into four segments: ascending aorta, aortic arch, descending aorta, and abdominal aorta, all of which were analyzed separately. A calcium score was then generated based on the degree of calcium detected in each of the seven distinct segments. Compared to controls, both study groups (root dilation and type A dissection) had lower total calcification across all 7 locations, with average combined calcification scores of 6.73 and 6.34, respectively, whereas the control group manifested more abundant calcium in both the aorta and coronary arteries, with an average combined calcification score of 9.36. Importantly, patients with type A dissections had the strongest negative association with atherosclerosis, displaying calcium scores that were 3.7 points lower than those of the control group (Figure 6). These findings were independent of known major risk factors for atherosclerosis, including BMI, gender, hypertension, diabetes mellitus, tobacco or illicit drug use, and dyslipidemia. Every analyzed major risk factor for atherosclerosis was associated with an increased risk of calcification. These results further emphasize the protective effect seen with aortic root aneurysms or ascending aortic dissections.

Our findings are consonant with prior work that similarly showed a lower incidence of coronary artery calcification among patients with dilated ascending thoracic compared to abdominal aortas [81,82]. Additionally, autopsy histology similarly reflects the findings that type A dissections have a lower incidence of atherosclerosis compared to both type B dissections and abdominal aortic aneurysms, expanding the body of literature that points to discrepant disease processes for the ascending and abdominal aortas [83,84].

### 3.4. CAD/Total Protection from MI

A significant clinical endpoint of atherosclerosis is coronary artery disease and myocardial infarction. The positive relationship between abdominal aortic aneurysms and coronary artery disease is well known [85,86,87,88]. Multiple studies have investigated whether this association extends to ascending thoracic aortic aneurysms.

Dr. Chau et al. from our team investigated the relationship between ascending thoracic aortic aneurysms and myocardial infarction to determine whether the protective effect of ATAAs on atherosclerosis extended to coronary artery disease as well [89]. In this study, 487 patients with ATAAs were compared to 500 control patients. Among this cohort, the aneurysmal group consisted of statistically significantly more patients with diabetes, whereas the patients in the control group had statistically significantly higher BMIs. Notably, patients with ATAAs showed dramatically reduced prevalence of MI with an odds ratio of 0.05, after controlling for age, gender, race, BMI, family history, hypertension, dyslipidemia, diabetes mellitus, and tobacco smoking (Figure 7 and Figure 8). Importantly, patients with ATAAs had higher BMIs than control patients, which is well known to be a risk factor for MI; still, they were protected despite the virulent BMI risk factor. Unsurprisingly, all of the risk factors for having a MI were associated with increased odds of experiencing a MI. Additionally, this study did a multivariable binary logistic regression which showed that patients with ATAAs were 298, 250, and 232 times less likely to suffer myocardial infarction than if they had a family history of MI, dyslipidemia, or hypertension, respectively. Thus, this study provides very strong evidence for the protective effect of ATAAs against myocardial infarctions.

Several additional studies assessed whether thoracic aneurysms resulted in increased cardiovascular risk and found, similarly to our study, that unlike abdominal aortic aneurysms, ascending thoracic aortic aneurysms have a notably diminished association with coronary artery disease or atherosclerosis [55,90,91,92]. Another study bolsters this body of evidence, finding, via histology, that patients with type A aortic dissections show extremely limited incidence of atherosclerotic lesions compared to controls [93]. Another study further supports these findings, showing that type B dissections have more severe atherosclerosis, more complications from ischemic heart disease, and more stenoses than type A dissections [94].

## 4. Aortic Histology

The absence of atherosclerotic plaques in ATAAs has also been previously shown in histological studies, with aortas from patients with ascending thoracic aneurysms having predominantly degenerative, non-atherosclerotic histopathology [95]. Another recent study compared the prevalence of atherosclerotic aortic lesions in patients with type A dissections versus controls, finding an extremely low incidence of lesions in dissection patients, whereas controls manifested abundant atherosclerotic lesions [93]. This paper validates an earlier study that showed similar findings of overwhelmingly medial degenerative changes to the aortic specimens from type A dissections [96]. However, an earlier histological study analyzed non-familial spontaneous ascending aortic aneurysms and found opposing results, with the majority of ATAAs showing atherosclerotic changes [97]. However, unlike the previous studies, this study follows an older classification of atherosclerosis and implements highly exclusionary criteria for the aneurysmal group, omitting patients with a known genetic connective tissue disease, a first-degree relative with thoracic aortic aneurysm or dissection, and who are aged ≤30 years. Another recent paper compared the ascending thoracic aortas of patients with aneurysms or dissections, again confirming the sole predominant histological finding of non-atherosclerotic medial degeneration that characterizes ascending aortic disease [98]. Thus, an intensive inspection of the microscopic histology of ascending thoracic aortic aneurysms provides further support for their protective role against atherosclerosis.

## 5. Discussion: Potential Mechanisms of Anti-Atherogenic Protection

There are several potential molecular mechanisms behind the observed pro-aneurysmal and anti-atherosclerotic effects of ascending thoracic aneurysms that have previously been proposed in the literature. A primary suggested mechanism involves the phenotypical switch defect of VSMCs and related matrix metalloproteinase (MMP) and transforming growth factor-β (TGF-*β*) pathways.

### 5.1. Aortic Vascular Smooth Muscle Cells

The thoracic aortic wall is populated by VSMCs, embryonically derived from the neural crest and second heart field cells. VSMCs, which play a crucial role in the maintenance of vascular wall homeostasis, can be divided into differentiated contractile and dedifferentiated synthetic phenotypes. In the adult aorta, VSMCs are mainly contractile to regulate vessel tone and luminal pressure and exhibit an extremely low proliferation rate. Unlike other differentiated cell types, VSMCs retain remarkable plasticity to undergo reversible changes in phenotype as a response to local environmental cues [99]. Stimulating or damaging factors such as vascular injury or growth factors induce VSMCs to switch from their differentiated, contractile phenotype to a dedifferentiated synthetic phenotype. The synthetic VSMCs express decreased levels of contractile and increased levels of synthetic proteins, high migratory capacity and proliferation rate, and massive production of the extracellular matrix [99,100].

Recent studies have shown that besides a transition of differentiation states, VSMCs can also take on the features of other cell types, such as osteoblasts, chondrocytes, adipocytes, and macrophage foam cells during the process of phenotypic switching [101]. Phenotypic switch of VSMCs therefore plays a key role in a number of human diseases [99]. The atherosclerotic disease process, for instance, may involve several stages—from adaptive intimal thickening to foam cell creation and migration and fibrous cap formation—in which VSMCs are involved. During the whole process of atherosclerotic lesion formation, not only are VSMCs transdifferentiated from a contractile to a synthetic VSMC phenotype, but also 70% of the cellular composition of the atherosclerotic plaque comes to consist of macrophage-like cells and osteochondrocytes, transdifferentiated from VSMCs [102].

### 5.2. Phenotypic Switch Defect in Thoracic Aortopathy Discourages Atherosclerosis

Although phenotypic switching is pivotal in several diseases, patient populations with an inherent susceptibility for thoracic aneurysm formation, such as in bicuspid aortic valve or Marfan Syndrome, showcase a defect in the ability to switch the VSMC phenotype [103,104]. Our previous work has shown that the ascending aortic wall in bicuspid aortic valve patients and Marfan patients, both non-dilated and dilated, is characterized by dedifferentiated VSMCs. Moreover, the intrinsic histological and morphological abnormalities are already apparent in the pediatric thoracic aorta in bicuspid aortic valve patients [105]. These findings suggest that the predominance of dedifferentiated VSMCs in thoracic aortopathy patients is not due to a pathology-induced phenotypic switch, but rather is embryonically determined. The inability of these VSMCs to switch phenotypes during their lifespan decreases vascular contractility and elasticity, leading to vessel wall dilatation. On the other hand, the lack of differentiation of VSMCs could protect against several diseases in which phenotypic switching is crucial, such as atherosclerosis.

### 5.3. MMP/TMP Dysregulation Accompanies VSMC Disturbances

Matrix metalloproteinases are a family of proteins that regulate the degeneration of collagen and elastin [106,107]. MMPs are inhibited by tissue inhibitors of metalloproteinases (TIMPs), and dysregulation of the balance between MMPs and TIMPs can result in the destruction of the extracellular matrix (ECM) [106,107,108,109,110,111,112]. Synthetic VSMCs can thus alter the surrounding matrix by the production and secretion of MMPs and their inhibitors. In line with a predominance of synthetic VSMCs in ATAAs, several studies have shown an elevation of plasma levels of MMPs in this patient population [113]. Additionally, MMPs have been found to be elevated in the thoracic aortic wall of patients with ATAAs and aortic dissection when compared to controls [114,115,116] as well as in aortic smooth muscle cells in the intima and media of patients with ATAAs [117].

Several studies have shown increased levels of MMP-2 in the aortas of patients with ATAAs [118] compared to patients with CAD [119,120]. Expression of MMP-3 in the ascending thoracic aortic wall has been shown to correlate inversely to the elasticity of the wall, with reduced elasticity potentially leading to aneurysm formation [121]. One meta-analysis showed increased MMP-9 and decreased TIMP1 and TIMP2 in the aortas of patients with ATAAs [122]. Additional studies showed increased MMP-9 levels in the media and adventitia of ATAA walls compared to controls, who showed low or absent levels [123]. MMP-12 has also been shown to be elevated in patients with ATAAs compared to controls [124,125].

TIMPs inhibit MMPs, and a previous gene expression profiling study showed decreased TIMP4 in aortic specimens of patients with type A aortic dissections [126], which would result in elevated MMP levels and elastin and collagen degradation in the ascending aortic wall.

MMPs have also been shown to be elevated in AAAs, with their collagenase activity destabilizing atherosclerotic plaques [127,128,129]. Plasmin has been shown to induce the activity of MMPs, resulting in the degradation of collagen and elastin [130]. AAAs specifically contain high levels of MMP-2, MMP-9, and MMP-12, whose high-elastase activity exposes components of the ECM that promote inflammatory macrophage infiltration [127,131,132,133,134,135,136,137,138]. This has been shown by an observed reduction in elastin and glycosaminoglycan content of AAA walls compared to controls [139]. MMP-2 in particular is known to promote atherosclerosis [140], and increased activity of MMPs has been shown to cause neointimal arterial lesions and smooth muscle cell migration [141,142,143,144].

Thus, the current literature shows that the imbalance of synthetic VSMCs in the vascular wall of thoracic aortopathy patients leads to an overproduction of matrix and degrading enzymes. Whereas the mucoid extracellular matrix accumulation and degeneration of elastin and collagen weakens the vascular wall and increases the risk for aortic complications, the lack of VSMC differentiation potential likely contributes to reducing the risk for atherosclerosis in the ascending aorta.

### 5.4. TGF-β Provides Anti-Atherogenic Contribution

TGF-*β* plays a pivotal role in the development and homeostasis of many organs, including the vasculature [145]. Mutations in TGF-*β* disrupt these processes, resulting in syndromes such as Marfan and Loeys–Dietz as well as potentially other thoracic aortic aneurysm proclivities [146]. A handful of evidence suggests that elevated TGF-*β* may be aortopathic, a concept supported by the increased TGF-*β* levels found in surgically removed aneurysmal aortas, as well as by animal studies showing a reduction in aortic dilatation via neutralization of TGF-*β* [147,148,149,150,151]. However, a wealth of studies shows opposing data, ranging from no change upon administration of neutralizing antibodies to increased aneurysmal expansion and rupture [152,153]. Additionally, multiple studies show that the deletion of TGF-*β* in mouse models leads to aortopathies [154,155,156,157].

Several studies have also shown marked differences in cellular responses to TGF-*β* between thoracic and abdominal aortas, including collagen production, collagen contractility, and extracellular (ECM) homeostasis [158,159,160]. Clinically, TGF-*β* has been shown to have pro-aneurysmal and anti-atherosclerotic effects [161,162,163,164]. Additionally, not only has TGF-*β* been shown to protect against the growth of atherosclerotic lesions, but it may also protect against plaque rupture [165,166,167]. Further, several studies have shown that some patients who suffer from MI have polymorphisms that decrease their levels of TGF-*β* [168,169,170,171,172,173,174].

These findings remain challenging to interpret as TGF-*β* has multiple remodeling properties, and an altered bioavailability in the serum cannot be identified as specific for aortic pathology or atherosclerosis. Additionally, isolating the exact role of TGF-*β* and its downstream signaling in aortic disease remains difficult because of its range of multiple age- and time-dependent phenotypes and the compensatory role of multiple TGF-*β* isoforms. However, both murine and human models have demonstrated that the absence of TGF-*β* activity decreases the VSMC contractility [175]. These findings are consistent with the decreased TGF-*β* activity seen in the bicuspid aortic valve and Marfan aorta. Furthermore, TGF-*β* plays a key role in the development of the intimal layer, and all patients with thoracic aortic pathology are characterized by a lack of intimal thickening [176]. Considering this essential role of TGF-*β* in the development of VSMCs and the intimal layer, it is plausible that in patients with a thoracic aortic pathology, a defect in the TGF-*β* signaling pathway contributes to the phenotypic switch defect and the anti-atherogenic absence of intimal thickening after birth.

Although a clearly defined mechanism is still needed, the current literature supports TGF-*β* being protective against atherosclerosis by contributing to the phenotypic switch defect, and as a predominantly protective anti-atherogenic.

### 5.5. Hemodynamic Changes

Alterations of the aortic hemodynamics have been shown to play a role in increasing the vascular susceptibility to the development of atherosclerosis. Specifically, the vascular endothelial lining is highly sensitive to variations in hemodynamic shear stress in the direction of blood flow. Blood flow is known to regulate vascular tone and structure through dynamic changes in wall shear stress, which result in the mechanically stimulated release of endothelial-derived factors such as prostaglandins, nitrovasodilators, and growth factors [177,178,179,180]. Additionally, it is known that certain geometries such as curves (aortic arch), branches, and bifurcations alter the flow in characteristic ways including turbulence and oscillatory shear stress that promote the focal development of atherosclerotic lesions [181,182,183,184]. This process then can perpetuate itself, as developing lesions further alter the endothelial shear stress pattern locally and promote further growth of that lesion. Furthermore, endothelial gene expression has been shown to respond to changes in local shear stress, with the potential to shift between atherosclerotic-protective and atherosclerotic-susceptible [185]. Differential transcriptome analyses have further refined this by demonstrating that shear stress alone combined with additional risk factors is necessary for changes in gene expression towards atherosclerotic susceptibility [186].

Pulse wave velocity (PWV) is the most widely used measure of arterial stiffness [187]. Increased PWV is associated with arterial atherosclerosis, which may be due to increased pulse pressure, leading to the formation and rupture of an atherosclerotic plaque [188]. Additionally, increased blood pressure is associated with adverse vascular remodeling [189]. Importantly, investigations in young patients (under 50 years old) with cardiovascular disease have shown increased PWV in the aortic arch [190]. Thus, variations in hemodynamics along the aorta play a key role in the development of atherosclerosis.

## 6. Conclusions/Future Steps/Limitations

As we see, a thorough body of evidence suggests that the underlying mechanisms of ATAAs protect against atherosclerosis, with potential mechanisms including impaired VSMC phenotypic switching and changes in the expression and levels of MMPs and TGF-*β*. These studies showing anti-atherosclerotic protection span carotid, coronary, and aortic imaging, as well as aortic histology and lipid homeostasis. These sources contribute a vast amount of supportive data manifesting and confirming the anti-atherosclerotic effects of ATAAs.

However, our understanding of the mechanisms of such protection remains elusive. This constitutes a major limitation of the work conducted to date. We speculate the dedifferentiated VSMCs and MMP and TGF-*β* pathways may be involved, but the true anti-atherogenic mechanisms in the ATAA setting remain to be clarified. Elucidation of these mechanisms may well add to our fundamental understanding of atherosclerosis, as well as lead to potential novel therapies for both ascending aortic aneurysm disease and even atherosclerosis itself. If we could therapeutically stimulate whatever pathways produce dramatic anti-atherogenic protection in ATAA patients, it is conceivable that this could beneficially impact the scourge of atherosclerotic disease on the human population.

## Figures and Tables

**Figure 1 ijms-24-15640-f001:**
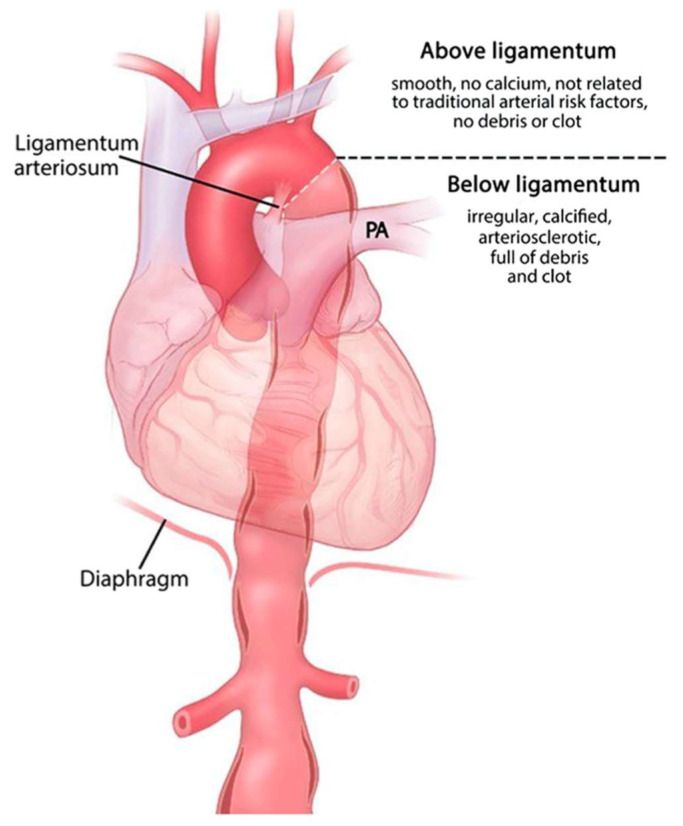
The thoracic aorta can be thought of as 2 separate organs when it comes to aneurysm disease. Above the ligamentum arteriosum, the aorta is nonatherosclerotic, noncalcified, and smooth in contour, whereas below the ligamentum, aortic aneurysms demonstrate heavy arteriosclerosis and calcification. PA = Pulmonary artery. Reproduced with permission from Farkas et al. [12].

**Figure 2 ijms-24-15640-f002:**
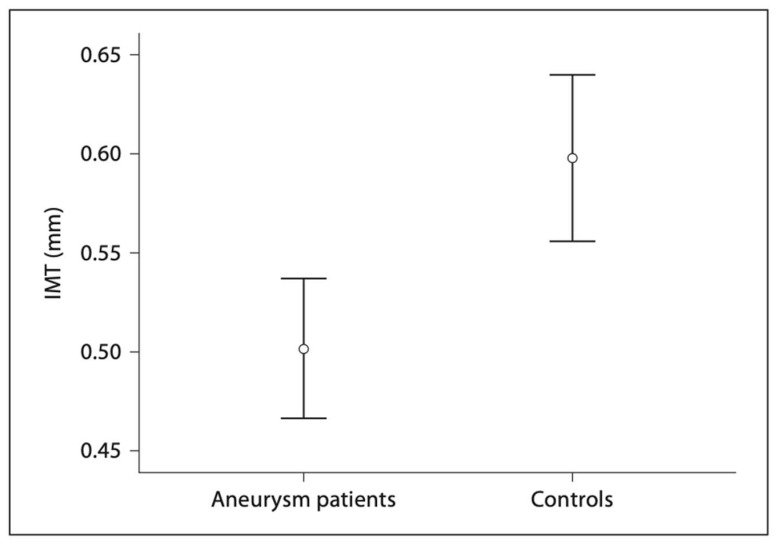
Average IMT (mm) for ATAA patients and controls with 95% confidence intervals indicated. Average IMT for aneurysm patients = 0.50 ± 0.13 mm. Average IMT for controls = 0.60 ± 0.11 mm. Difference is statistically significant (*p* < 0.01). Reproduced with permission from Hung et al. [40].

**Figure 3 ijms-24-15640-f003:**
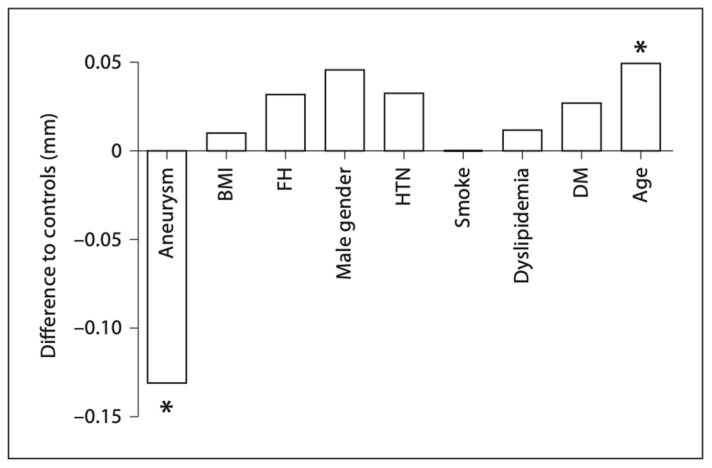
Difference in overall IMT scores (regression coefficient β) relative to the control group. Age is depicted as a 10-year interval. BMI is depicted as a 10-unit interval. DM = diabetes mellitus; Smoke = smoking. * *p* < 0.01 is statistically significant. Reproduced with permission from Hung et al. [40].

**Figure 4 ijms-24-15640-f004:**
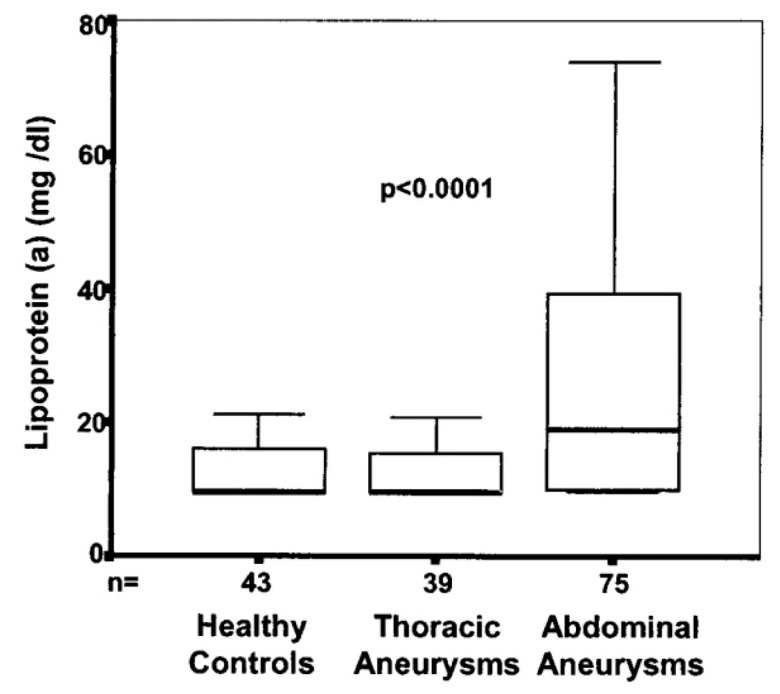
Serum levels of lipoprotein (a) in patients with AAA (n = 75), thoracic aortic disease (n = 39), and healthy control subjects (n = 43). Box plots indicate median, IQR (range from 25th to 75th percentile), and range. Reproduced with permission from Schillinger et al. [43].

**Figure 5 ijms-24-15640-f005:**
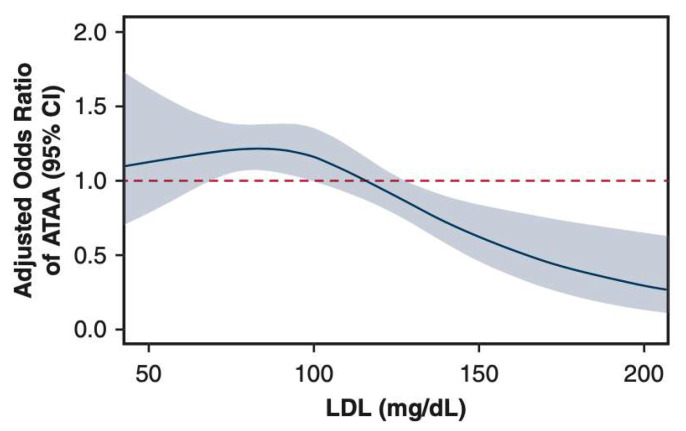
Adjusted cubic spline for the conditional odds of ATAA by low-density lipoprotein (LDL) levels. The spline model was adjusted for age, sex, smoking status, statin use, diabetes mellitus, hypertension, and body mass index. Shaded region represents a 95% confidence interval (CI). Reproduced with permission from Weininger et al. [51].

**Figure 6 ijms-24-15640-f006:**
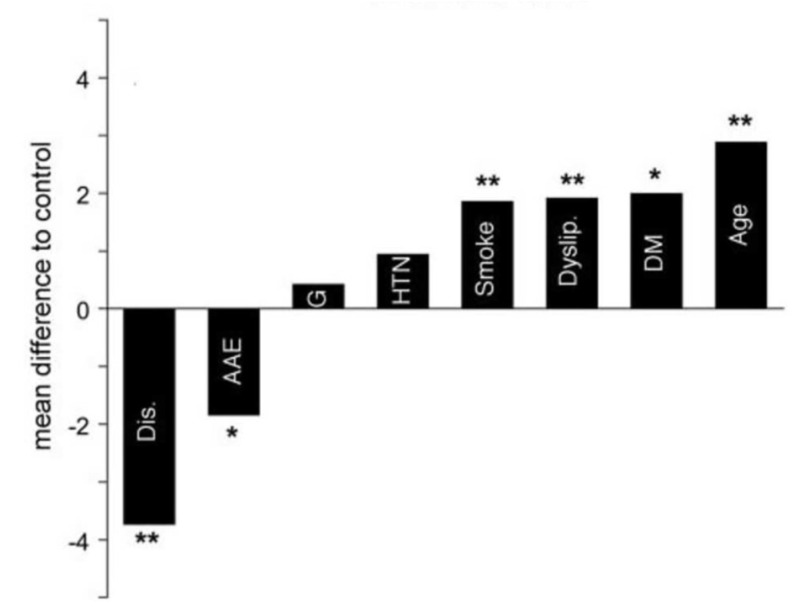
Difference in overall calcification scores (regression coefficient β) relative to the control group, for all risk factors analyzed. Age is depicted as a 10-year interval. AAE = annuloaortic ectasia; Age = age in 10-year intervals; Dis. = dissection; DM = diabetes mellitus; Dyslip. = dyslipidemia; G = male gender; HTN = hypertension; Smoke = smoking; * = statistically significant with *p* < 0.5; ** = statistically significant with *p* < 0.01. Reproduced with permission from Achneck et al. [80].

**Figure 7 ijms-24-15640-f007:**
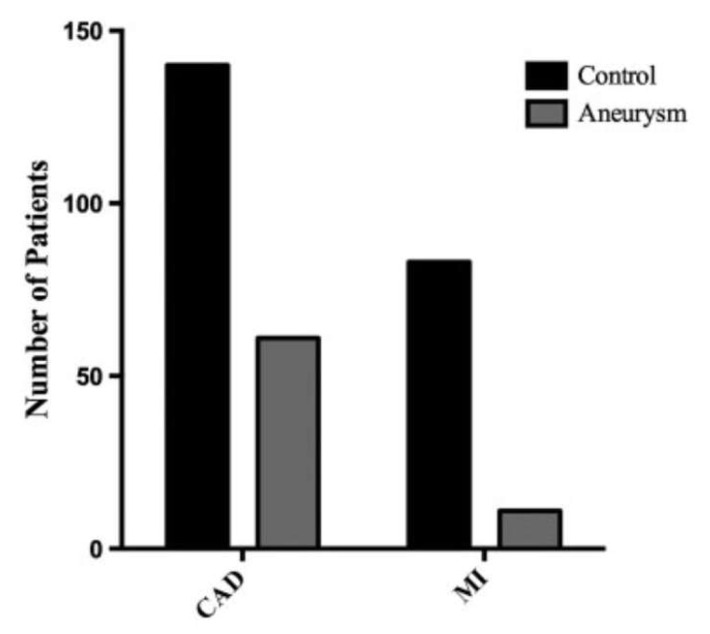
There was a significantly lower prevalence of CAD and MIs in the aneurysm group compared to the control group. There were 61 patients in the aneurysm group who had CAD (*p* < 0.001) versus 140 in the control group. There were 11 patients in the aneurysm group who have had a MI versus 83 in the control group (*p* < 0.001). CAD = coronary artery disease; MI = myocardial infarction. Reproduced with permission from Chau et al. [89].

**Figure 8 ijms-24-15640-f008:**
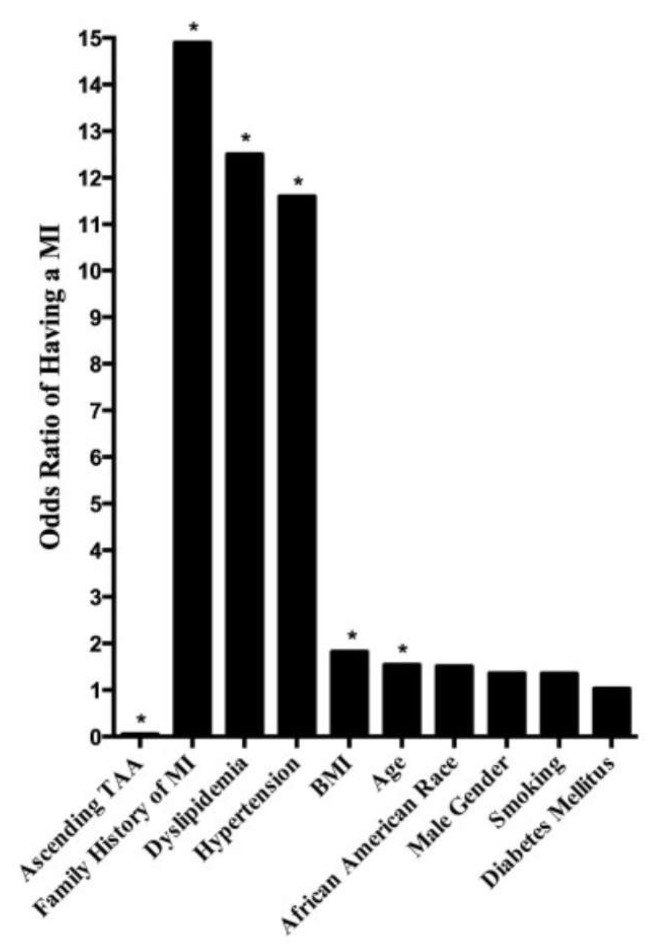
The odds ratio of having a MI when a patient had an ATAA versus MI risk factors. Age is in 10-year intervals, and BMI is in 10-unit intervals. Note the drastic difference in odds ratios when a patient had an ATAA compared with when a patient had a MI risk factor. * Statistically significant (*p* < 0.05). BMI = body mass index; MI = myocardial infarction; TAA = thoracic aortic aneurysm. Reproduced with permission from Chau et al. [89].

## Data Availability

No new data was created or analyzed in this study. Data sharing is not applicable to this article.

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
