# Peer review of "Evidence Accumulates: Patients with Ascending Aneurysms Are Strongly Protected from Atherosclerotic Disease"

_ijms, 2023, doi:10.3390/ijms242115640_

Round 1
Reviewer 1 Report
This manuscript details a compelling body of work suggesting that ascending thoracic aortic aneurysms may offer a 'silver lining' of anti-atherogenic protection through a number of pathophysiologic mechanisms.
The content is comprehensive and will be of great interest to a varied and multidisciplinary audience, and the justifications are presented elegantly.
There are >150 references included in the preparation of this work, and my only suggestion for consideration by the authors might be to include some additional figures from external sources for balance. Although a vast majority of this research has originated in the senior author's primary institution, including additional external data in tabular / graphic / pictorial form, whether it be supportive or contrary to the current authors' findings, may add additional dimension to the manuscript.
Many thanks for the opportunity to review this excellent submission.
Reviewer 2 Report
The narrative review "Patients with Ascending Aneurysms are Strongly Protected from Atherosclerotic Disease "addresses the relationship between ascending thoracic aortic aneurysms (ATAA) and atherosclerosis. It focuses on the comparison of morphological features (calcification, histological incl. embryonical derivation and phenotypic switch), blood biomarkers from (LDL, HDL) and "molecular mechanisms" (e.g. TGFß) between patients with ATAAs (based in particular on the authors' own publications) and patients with atherosclerosis. The authors conclude that "ATAAs protect against atherosclerosis".
The topic is of general interest and contributes to understanding these disease groups' pathophysiology. However, I have some comments on the presentation of the issue.
Major points
When comparing two completely different vascular pathologies, one has to take a step back and name the determinants or causes of the respective diseases as far as known. While in the case of atherosclerosis, the traditional cardiovascular risk factors play a significant role in the review article and the underlying studies, potential (acquired) triggers of ATAA are hardly mentioned. Here are just two examples:
- Subsequent diagnosis of aortitis underlying ATAA (ref. example: Schmidt J, Sunesen K, Kornum JB, Duhaut P, Thomsen RW. Predictors for pathologically confirmed aortitis after resection of the ascending aorta: a 12-year Danish nationwide population-based cross-sectional study. Arthritis Res Ther. 2011 Jun 15;13(3):R87. doi: 10.1186/ar3360)
- 17-fold increased risk of developing ATAA in patients with giant cell arteritis (Evans JM, O'Fallon WM, Hunder GG. Increased incidence of aortic aneurysm and dissection in giant cell (temporal) arteritis. A population-based study. doi: 10.7326/0003-4819-122-7-199504010-00004)
Data concerning functional and haemodynamic changes in aortic diseases and atherosclerosis is still scarce but growing rapidly due to advances in medical technology. I recommend at least mentioning this topic. As an aid for the authors, only the keywords "pulse wave velocity "and "wall shear stress "should be mentioned here.
The authors rely on carotid IMT as a biomarker to detect early atherosclerosis. However, the value of carotid IMT by ultrasound examination for risk stratification remains controversial (Kim GH, Youn HJ. Is Carotid Artery Ultrasound Still Useful Method for Evaluation of Atherosclerosis? Korean Circ J. 2017 Jan;47(1):1-8. doi: 10.4070/kcj.2016.0232). The limitations of carotid IMT measurement should be mentioned for a balanced presentation.
Finally, the conclusion "ATAAs protect against atherosclerosis" is wrong. It cannot be deduced from the cited studies that ATAAs themselves protect against atherosclerosis; at best, the underlying mechanisms of ATAAs do. However, the title "Patients with Ascending Aneurysms are Strongly Protected…" seems correct.
Similarity: The lines 349 – 356 show a high similarity with the text of the following publication: Grewal N, Klautz R, Poelmann RE. Can transforming growth factor beta and downstream signalers distinguish bicuspid aortic valve patients susceptible for future aortic complications? Cardiovasc Pathol. 2023 Mar-Apr;63:107498. doi: 10.1016/j.carpath.2022.107498. Epub 2022 Nov 18. PMID: 36403918.
Minor:
The text gives the "13-18% decrease in risk of stroke and a 10-15% decrease in risk of myocardial infarction "as ref. 33, elsewhere as ref. 26. Please check.
Line 123: "As a meaningful marker and risk factor for vascular disease, LDL can may provide further insight ". Please correct the sentence structure.
Please correct punctuation and spaces, e.g. in line 78.
Round 2
Reviewer 2 Report
I propose the following changes in the text marked in red. (Lines 35-37):
“Our work and that of many other teams has demonstrated that many ascending aortic aneurysms are often familial in origin and genetically triggered. Among the minority of patients without genetic etiology, inflammation and infection are known, non-atherosclerotic triggers.”
